# Block-Recurrent Transformers

**DeLesley Hutchins**[*1], **Imanol Schlag**[*3†], **Yuhuai Wu**[1], **Ethan Dyer**[2], **Behnam Neyshabur**[2]

[1] Google Research      [2] Google Research, Blueshift Team
[3] The Swiss AI Lab IDSIA, SUPSI & USI
{delesley, yuhuai, edyer, neyshabur}@google.com      imanol@idsia.ch

## Abstract

We introduce the Block-Recurrent Transformer, which applies a transformer layer in a recurrent fashion along a sequence, and has linear complexity with respect to sequence length. Our recurrent cell operates on blocks of tokens rather than single tokens during training, and leverages parallel computation within a block in order to make efficient use of accelerator hardware. The cell itself is strikingly simple. It is merely a transformer layer: it uses self-attention and cross-attention to efficiently compute a recurrent function over a large set of state vectors and tokens. Our design was inspired in part by LSTM cells, and it uses LSTM-style gates, but it scales the typical LSTM cell up by several orders of magnitude. Our implementation of recurrence has the same cost in both computation time and parameter count as a conventional transformer layer, but offers dramatically improved perplexity in language modeling tasks over very long sequences. Our model out-performs a long-range Transformer XL baseline by a wide margin, while running twice as fast. We demonstrate its effectiveness on PG19 (books), arXiv papers, and GitHub source code. Our code has been released as open source [1].

## 1 Introduction

Transformers have mostly replaced recurrent neural networks (RNNs), such as LSTMs [2], on tasks that involve sequential data, especially natural language. There are several reasons for their success. First, transformers process all elements of the sequence in parallel, and are thus faster to train on modern accelerator hardware. In contrast, an RNN must process tokens sequentially, which leads to slow step times during training, and large batch sizes in order to fully saturate GPUs or TPUs.

Second, an RNN must summarize and compress the entire previous sequence into a single state vector which is passed from one token to the next. The size of the state vector limits the amount of information that the RNN can encode about the previous tokens in the sequence. In contrast, a transformer can attend directly to past tokens, and does not suffer from this limitation.

Third, attention operates effectively over longer distances. The forget gate in an LSTM discards information moving forward, and causes vanishing gradients during backpropagation. In practice, this means that LSTMs struggle to send a clear signal over more than a few hundred tokens, far less than the typical size of the attention window in a transformer [3].

Despite these advantages, transformers also have a disadvantage. The computational complexity of self-attention is quadratic with respect to the sequence length, which is a limiting factor when attempting to process long documents, such as books, technical articles, or source code repositories. Moreover, a transformer has no memory of past context; any tokens that it cannot attend to are "invisible" to the model.

---

[*]Equal Contribution      † Work done partially while interning at Google Research (Blueshift Team) and partially funded by ERC Advanced grant no: 742870 to J.Schmidhuber.

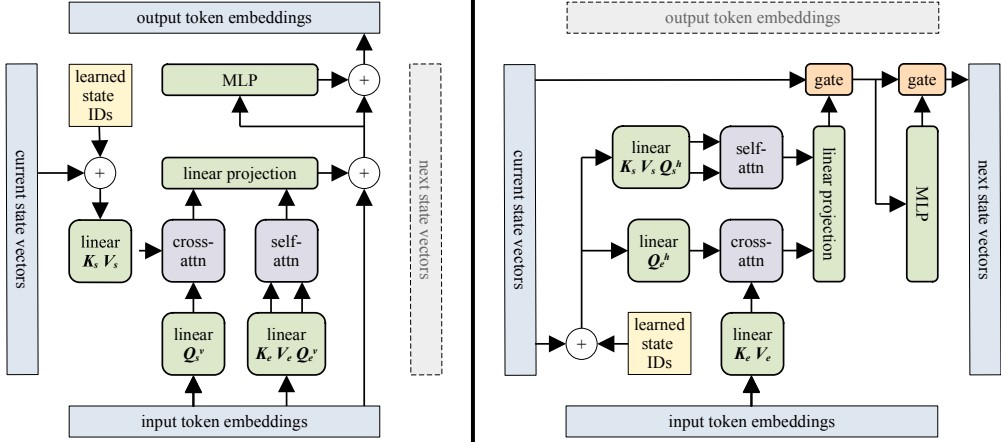

Figure 1: Illustration of our recurrent cell. The left side depicts the vertical direction (layers stacked in the usual way) and the right side depicts the horizontal direction (recurrence). Notice that the horizontal direction merely rotates a conventional transformer layer by 90°, and replaces the residual connections with gates.

In this work, we describe an architecture which combines the benefits of attention and recurrence. Like previous implementations of recurrence, our architecture constructs and maintains a fixed-size state, which summarizes the sequence that the model has seen thus far. However, our implementation of recurrence differs from previous work in several important aspects which together address the three limitations mentioned above.

Instead of processing the sequence one token at a time, **our recurrent cell operates on *blocks* of tokens**; see Figure 1. Within a block, all tokens are processed in parallel, at least during training. The recurrent cell likewise **operates on a *block* of state vectors rather than a single vector.** This means that the size of the recurrent state is orders of magnitude larger than in an LSTM, which dramatically improves the model's capacity to capture the past. Processing the sequence in blocks also helps propagate information and gradients over longer distances, because the number of recurrent steps (and thus the number of times that the forget gate is applied) is orders of magnitude smaller. We show that the Block-Recurrent Transformer can remember information over distances of 60k tokens or more.

The recurrent cell itself is strikingly simple. For the most part, it consists of an ordinary transformer layer applied in a recurrent fashion along the sequence length. There are **a few tricks that are necessary to stabilize training**; see Sections 3.2 and 3.4 for details. The cost of recurrence, in terms of both computation time and parameter count, is essentially the same as simply adding one more layer to our transformer baseline. We demonstrate empirically that adding a single recurrent layer results in a much larger improvement in perplexity on multiple datasets than adding a conventional transformer layer, while training time and memory use are equivalent. Moreover, our recurrent cell is very easy to implement because it largely makes use of existing transformer code. Thus, our technique is a cheap and cheerful way to improve language modeling perplexity on long sequences.

## 2 Related Work

The quadratic cost of attention is well known in the literature, and a great deal of work has been done on efficient long-range attention mechanisms; see [4, 5] for recent surveys. Sparse strategies such as Big Bird [6], Routing Transformers [7], and Reformer [8] select only a subset of tokens to attend to. Hierarchical mechanisms [9] combine multiple tokens into phrases or sentences to reduce sequence length. Expire-span [10] learns to prune far-away tokens that the model has labelled as "unimportant". Memorizing transformers [11] replace dense attention with $k$-nearest-neighbor lookup.

Yet another approach is to reduce the sequence length by pooling, averaging, or compressing it in some way. Hierarchical 1D attention [12], and Combiner [13] apply pooling or averaging over tokens at longer distances. Linformer [14] applies a linear transformation to the key and value matrices to reduce the sequence length. Compressive transformers [15] and funnel transformers [16] apply additional learned compression layers to compress the sequence.

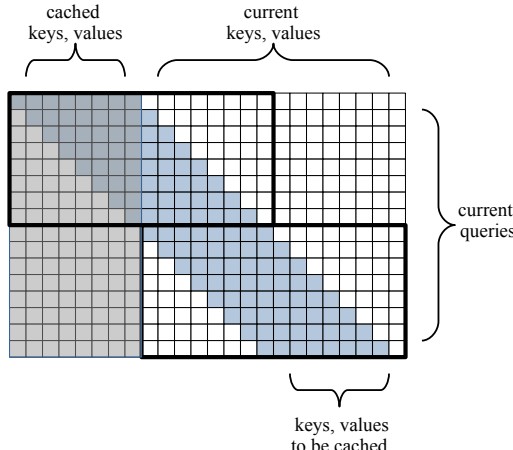

Figure 2: Sliding window, where segment length $N = 16$, window/block size $W = 8$. Keys and values for the first $W$ shaded tokens were computed and cached on the previous training step; the remaining $N$ unshaded tokens are the segment for the current training step. Instead of a single $N \times (W + N)$ attention matrix, attention is done in two tiles of size $W \times 2W$.

The equation for attention is (roughly) $\mathsf{softmax}(\boldsymbol{Q}\boldsymbol{K}^T)\boldsymbol{V}$ where $\boldsymbol{Q}$, $\boldsymbol{K}$, and $\boldsymbol{V}$ are the query, key, and value matrices of the attention layer. If the softmax operation is removed from this equation or somehow "linearized", the equation can be rearranged as $\boldsymbol{Q}(\boldsymbol{K}^T\boldsymbol{V})$, where $(\boldsymbol{K}^T\boldsymbol{V})$ can be computed incrementally (i.e., in a recurrent fashion) as a cumulative sum over the sequence [17]. Linearized attention thus has linear rather than quadratic complexity with respect to sequence length. Following this line of reasoning, there have been several proposals that approximate the softmax [18, 19] or replace it [20, 21]. Linear transformers are related to earlier work on fast weight programmers [20] [22], and can be extended with other forms of recurrence [23].

Our work differs from all of the above mechanisms, because we rely only on standard dense attention with softmax.

A few other lines of research have combined the transformer architecture with recurrence in some way. The feedback transformer [24] allows lower layers to attend to the output of the topmost layer. Feedback has minimal cost at inference time, but it is unfortunately very slow to train because tokens must be processed sequentially. Simple Recurrent Units [25, 26] use a recurrence function that does not involve matrix multiplication, and is consequently much faster. RNMT+ combines RNNs and transformers in an encoder/decoder architecture to improve on translation tasks [27]. "Sandwich models" alternate between transformer and RNN layers and out-perform both transformers and RNNs on tasks involving source code [28]. The R-Transformer introduces an additional local RNN which can be computed in parallel in order to better model sequential structure [29]. The Perceiver architecture [30] is somewhat similar to ours; it also applies a transformer layer in an iterative fashion.

To the best of our knowledge, the idea of performing recurrence on blocks of tokens is underexplored. In the context of translation, [31] operates on sentences rather than tokens. Staircase Attention [32] also operates on blocks of tokens; each layer takes, as input, the outputs of the same layer from the previous block.

## 3    Method

The Block-Recurrent Transformer is based on sliding-window attention [33], which is an extension of ideas from Transformer-XL [34].

A long document, such as a book, consists of a *sequence* of tokens. Due to memory limitations, it is usually not possible to fit the entire sequence into device memory. Thus, the sequence is divided into *segments* of length $N$ ($N = 4096$ in our experiments), which are processed sequentially over a number of training steps. Each training step processes one segment.

The sliding window attention pattern is illustrated in Figure 2. Given a segment of $N$ tokens, the sliding window applies a causal mask in which each token can only attend to the $W$ previous tokens,

where $W$ is the *window size* ($W = 512$ in our experiments). Because of the causal mask, most entries of the $N \times N$ attention matrix are masked out (assuming that $W << N$). Thus, the attention computation can be optimized by breaking it into smaller tiles along the diagonal. The segment of $N$ tokens is subdivided into *blocks* of size $W$, and each block attends locally to itself and to the previous block, so the size of each local attention matrix is $W \times 2W$. Using this mechanism, attention is quadratic with respect to the window size $W$, but linear with respect to the segment length $N$.

Borrowing an idea from Transformer-XL, the keys and values from the last block in each segment are stored in a non-differentiable *cache* for use on the next training step. By using the cache, the first block in the next segment can attend to the last block in the previous segment, which extends the sliding window to cover the entire (book-length) sequence. The cache implements a form of truncated backpropagation through time [35] over long documents.

Note that if $N = W$, then sliding window attention will behave exactly like Transformer-XL; it will process and cache one segment (i.e. one block) per training step. Setting $N >> W$ does not change the context length of attention, but it allows gradients to backpropagate across multiple blocks during training; we show that the improved differentiability provides a modest benefit to perplexity over Transformer-XL. See Appendix A for more details.

## 3.1 Recurrent Cell

A Block-Recurrent Transformer layer extends the sliding-window attention mechanism by adding a set of recurrent states, which are updated at the end of each block of $W$ tokens. Our design for the recurrent cell is illustrated in Figure 1, which depicts the operations done within a single block of the input sequence.

The recurrent cell receives two tensors as inputs: a set of $W$ token embeddings, where $W$ is the block/window size, and a set of $S$ "current state" vectors. The cell produces two tensors as outputs: a set of $W$ output embeddings, as well as a set of $S$ "next state" vectors. We denote the function going from input token embeddings to output token embeddings as the *vertical* direction, and the function going from the current state vectors to the next state vectors as the *horizontal* direction. The number of state vectors $S$ and the window size $W$ are independent hyperparameters, but we set $S = W = 512$ in our experiments to simplify comparisons against baselines.

The **vertical direction** of the cell is an ordinary transformer layer with an additional cross-attention operation, much like a decoder layer in a standard encoder-decoder architecture [36]. It does self-attention over the input tokens, and cross-attends to the recurrent states. Unlike a typical decoder layer, we do self-attention and cross-attention in parallel. The results of both forms of attention are concatenated together and fed into a linear projection.

The **horizontal direction** of the cell mirrors the forward direction, except that it performs self-attention over the current state vectors, and cross-attends to the input tokens. The recurrent direction also replaces the residual connections with gates, which allows the model to "forget", an ability that is important for algorithmic tasks [37], or when processing long documents, where it has been central to the success of LSTMs [38].

Note that the presence of gates is the reason why self-attention and cross-attention are done in parallel. Doing them sequentially, as is standard practice, would introduce a third gate in the horizontal direction, which led to worse perplexity in our experiments.

Recurrence is integrated with the sliding window attention mechanism. Although not shown in Figure 1, each cell also receives keys and values from the previous block as input, these are concatenated with $(\boldsymbol{K}_e, \boldsymbol{V}_e)$ from the current block in order to implement sliding-window attention.

A Block-Recurrent Transformer *layer* processes the blocks within a segment sequentially by stacking recurrent cells horizontally, with the "next states" output of the previous cell feeding into the "current states" input of the next cell. In code, this is implemented as a simple for-loop over blocks. Multiple layers can also be stacked vertically in the usual fashion. Our experiments use a single recurrent layer, sandwiched between a number of non-recurrent layers that use sliding-window attention.

The final set of state vectors from the last block in the segment are cached, along with the keys and values, and used as the initial state for the first block on the next training step. Every layer in the stack (both recurrent and non-recurrent) has its own cache.

**Sharing of keys and values.** Keys and values are shared between the vertical and horizontal directions. One set of keys and values $(\boldsymbol{K}_e, \boldsymbol{V}_e)$ are computed from the input token embeddings, and another set of keys and values $(\boldsymbol{K}_s, \boldsymbol{V}_s)$ are computed from the recurrent state vectors. Queries are not shared, so there are four separate sets of queries: $\boldsymbol{Q}_e^v$ and $\boldsymbol{Q}_s^v$ in the vertical direction, and $\boldsymbol{Q}_s^h$ and $\boldsymbol{Q}_e^h$ in the horizontal direction.

### 3.2 State IDs and Position Bias

With a large number of state vectors, the total size of the recurrent state is far larger than that of an LSTM. However, the same weights (projection matrices and MLP) are applied to each state vector. Without some way to differentiate the states, the model will compute the same result for each state vector, thus negating any advantage from having multiple states. To prevent this failure mode, we add a set of learned "state IDs" to the state vectors before computing the keys, values, and queries. These "state IDs" allow each state vector to consistently issue different queries against the input sequence, and against other states. State IDs are identical to learned position embeddings; we use a different name because there's no notion of "position" between states.

We do not add global position embeddings to the tokens, because global position embeddings don't work well for long sequences [34]. Instead, we add a T5-style relative position bias [39] to the self-attention matrix in the vertical direction. (Although similar, T5 relative positions differ slightly from the relative positions used in the Transformer-XL paper [34].) When the recurrent states cross-attend to input tokens, there is no position bias, because the relative distance between "state" and "token" is undefined.

We also normalize queries and keys as described in [40]; we found that normalization improved the stability of Transformer-XL when used with a relative position bias.

### 3.3 Gate Type

We experimented with two different gating mechanisms for the recurrent cell. Each state vector has its own gate, but all state vectors are updated in parallel, using the equations below.

**Fixed gate.** The fixed gate uses a learned convex combination, similar to highway networks [41].

$$\boldsymbol{z}_t = \boldsymbol{W}_z \boldsymbol{h}_t + \boldsymbol{b}_z \tag{1}$$

$$\boldsymbol{g} = \sigma(\boldsymbol{b}_g) \tag{2}$$

$$\boldsymbol{c}_{t+1} = \boldsymbol{c}_t \odot \boldsymbol{g} + \boldsymbol{z}_t \odot (1 - \boldsymbol{g}) \tag{3}$$

where $\boldsymbol{W}_z$ is a trainable weight matrix, $\boldsymbol{b}_z$ and $\boldsymbol{b}_g$ are trainable bias vectors, $\sigma$ is the sigmoid function, $\boldsymbol{c}_t$ is the cell state for the current block (i.e., the state for the block at index $t$ in the sequence of blocks), $\odot$ is the element-wise multiplication, and $\boldsymbol{h}_t$ is the current input to the gate. In our model, $\boldsymbol{h}_t$ is either the output of attention, in which case $\boldsymbol{W}_z$ is the linear projection that feeds into the gate, or $\boldsymbol{h}_t$ is the output of the hidden layer of the MLP, in which case $\boldsymbol{W}_z$ is the final layer of the MLP.

Unlike highway networks, the bias $\boldsymbol{b}_g$ is a simple learned vector of shape $\mathbb{R}^d$, which is broadcast over all state vectors, where $d$ is the state embedding dimension. The value of $\boldsymbol{g}$ does **not** depend on either the current value of the state vector $\boldsymbol{c}_t$, or on the current input $\boldsymbol{h}_t$, and thus remains constant (i.e., fixed) after training. The fixed gate essentially implements an exponential moving average over previous blocks.

**LSTM gate.** The LSTM gate uses the standard combination of input and forget gates:

$$\boldsymbol{z}_t = \tanh(\boldsymbol{W}_z \boldsymbol{h}_t + \boldsymbol{b}_z) \tag{4}$$

$$\boldsymbol{i}_t = \sigma(\boldsymbol{W}_i \boldsymbol{h}_t + \boldsymbol{b}_i - 1) \tag{5}$$

$$\boldsymbol{f}_t = \sigma(\boldsymbol{W}_f \boldsymbol{h}_t + \boldsymbol{b}_f + 1) \tag{6}$$

$$\boldsymbol{c}_{t+1} = \boldsymbol{c}_t \odot \boldsymbol{f}_t + \boldsymbol{z}_t \odot \boldsymbol{i}_t \tag{7}$$

where $\boldsymbol{W}_z, \boldsymbol{W}_i, \boldsymbol{W}_f$ are trainable weight matrices, and $\boldsymbol{b}_z, \boldsymbol{b}_i, \boldsymbol{b}_f$ are trainable bias vectors. The LSTM gate is strictly more expressive, because the values of $\boldsymbol{f}_t$ and $\boldsymbol{i}_t$ depend on the current input $\boldsymbol{h}_t$. In our model, $\boldsymbol{h}_t$ depends on $\boldsymbol{c}_t$, so the LSTM gate also depends indirectly on $\boldsymbol{c}_t$. LSTM gate values are thus different for each state vector, and for each block index $t$.

### 3.4 Gate Initialization and Training Stability

We observed that training stability is quite sensitive to how the gates are initialized. Recurrence has a failure mode where the model learns to completely ignore the recurrent state, in which case its performance reverts to that of the non-recurrent transformer. Moreover, this situation appears to be a local optimum; once the model has reached this point, it does not recover. We stabilize training by initializing the weights and bias to small but non-zero values, and adding a constant -1 and +1 to the input and forget gates to bias them to "remember". See Appendix B for details.

### 3.5 Gate Configuration

We experimented with three different gate configurations.

**Dual.** The dual gate configuration is the one shown in Figure 1, in which both of the residual connections in the cell are replaced with gates. The disadvantage of this configuration is that there are two gates, both of which can forget.

**Single.** The single gate configuration removes the linear projection and the gate that is attached to it. Instead, the concatenation of self-attention and cross-attention is fed directly into the MLP.

**Skip.** The skip configuration removes the MLP and the gate that is attached to it. This configuration is similar to the single-gate version, except that it is strictly weaker. Instead of a two layer MLP with a very large hidden layer, it uses a linear projection with no nonlinearity.

### 3.6 Placement of Recurrence and Computation Cost

**Single recurrent layer.** The basic version of the Block-Recurrent Transformer uses a single recurrent layer sandwiched between a number of non-recurrent transformer layers with sliding attention. We use a 12-layer model with recurrence on layer 10. All layers have a Transformer-XL-style cache.

**Cost of recurrence.** During training, the 12-layer Block-Recurrent Transformer has almost exactly the same computation cost, in both parameters and FLOPS, as a 13-layer Transformer-XL model without recurrence. The two are equivalent because the recurrent cell does almost the same operations as a conventional transformer layer, merely in the horizontal instead of the vertical direction.

The inference cost for autoregressive decoding is also nearly identical, for the same reason. Recurrence adds an additional attention operation per token, the cost of which is the same as self-attention in a 13th layer.

## 4 Results

We tested the Block-Recurrent Transformer on three different data sets of long documents: PG19, arXiv, and GitHub. The PG19 dataset [42] contains full-length books written prior to 1919 from project Gutenberg. The arXiv dataset [11] is a corpus of technical papers downloaded via the arXiv Bulk Data Access[1], and filtered to include only articles labeled as "Mathematics" and whose LaTeX source is available. The GitHub dataset [11] is a corpus of source code from different GitHub repositories with open-source licenses. All of the files in each GitHub repository are concatenated together to make one long document.

The task is auto-regressive language modeling, where the goal is to predict the next token in the sequence. We report bits-per-token numbers (i.e. $\log_2$ perplexity; lower is better) for all models. Further training details for each dataset can be found in Appendix C.

### 4.1 Baselines

We compare the Block-Recurrent Transformer to five different baselines. The first baseline, XL:512, establishes a reference point against which various other improvements can be compared. It's a

---

[1] https://arxiv.com/help/bulk_data

Table 1: Average bits-per-token ($\log_2$ perplexity) of each model. The recurrent models (named `Rec:gate:config`) have the same computational cost as the `Slide:13L` baseline, but much better perplexity. They even outperform the `XL:2048` baseline, **while running more than twice as fast.** Measured error bars on PG19 are low, between 0.002 and 0.007, but are rounded up to 0.01 to match the precision of results in the table. Step time is for a single training step (lower is better). For PG19, we train both character-level (bytes) and token-level models.

| Model | segment length | window length | step time (relative) | PG19 bytes | PG19 tokens | arXiv tokens | GitHub tokens |
|---|---|---|---|---|---|---|---|
| XL:512 | 512 | 512 | 0.88 | 1.01 | $3.62 \pm 0.01$ | 1.45 | 1.21 |
| XL:1024 | 1024 | 1024 | 1.20 | 0.997 | $3.59 \pm 0.01$ | 1.37 | 1.08 |
| XL:2048 | 2048 | 2048 | 2.11 | 0.990 | $3.58 \pm 0.01$ | 1.31 | 1.01 |
| Slide:12L | 4096 | 512 | 0.93 | 0.989 | 3.60 | 1.43 | 1.19 |
| Slide:13L | | | 1.00 | 0.989 | $3.58 \pm 0.01$ | 1.42 | 1.17 |
| Rec:lstm:dual | 4096 | 512 | 1.06 | 0.985 | $3.54 \pm 0.01$ | 1.26 | 1.01 |
| Rec:lstm:single | | | 1.05 | 0.962 | $3.54 \pm 0.01$ | 1.29 | 1.03 |
| Rec:lstm:skip | | | 1.00 | 0.969 | $3.56 \pm 0.01$ | 1.31 | 1.10 |
| Rec:fixed:dual | | | 1.01 | 0.957 | $\mathbf{3.52 \pm 0.01}$ | 1.27 | 0.991 |
| Rec:fixed:single | | | 1.02 | 0.966 | $3.58 \pm 0.01$ | 1.25 | 1.00 |
| Rec:fixed:skip | | | **0.99** | **0.952** | $3.53 \pm 0.01$ | **1.24** | **0.976** |
| Feedback:lstm:single | 4096 | 512 | 1.40 | 0.977 | 3.50 | **1.22** | - |
| Feedback:fixed:skip | | | 1.35 | **0.935** | **3.49** | 1.24 | - |
| Memorizing Trans. 64k | 512 | 512 | 1.94 | 0.950 | 3.53 | **1.22** | - |

12-layer Transformer-XL model with a window size of 512, and 150 million parameters. It has 8 heads of size 128, embedding vectors of size 1024, an MLP with a hidden layer of size 4096, and the relu nonlinearity. It uses a Transformer-XL style cache, but no sliding window, so the segment length is the same as the window size, i.e., it is trained on segments of 512 tokens.

`XL:1024` and `XL:2048` are similar, but have window sizes of 1024 and 2048, respectively. As expected, increasing the window size improves perplexity, especially on the arXiv data set. However, these two models still have worse perplexity than the recurrent model, as well as being much slower.

`Slide:12L` is a 12-layer transformer with a window size of 512, but uses a sliding window over a segment of 4096 tokens. This model is almost identical to `XL:512`; the only difference is that the sliding window is differentiable over multiple blocks, while the Transformer-XL cache is not.

`Slide:13L` adds a 13th layer, and is directly comparable to the recurrent models in terms of both computation cost (FLOPS or step-time), number of parameters, and segment length. Notice that adding another layer with more parameters yields a much smaller improvement than adding recurrence.

**Relative cost.**   All five baselines, and all 6 recurrent models, have roughly the same number of parameters: between 151 million (12 layer) and 164 million (13 layer or recurrent). The training speed (i.e. step time) of each model is shown in Table 1 (lower is better). Because the raw step time depends on hardware and compiler, we report numbers relative to the `Slide:13L` baseline.

**Batch Size.**   We adjust the batch size so that each model processes the same number of tokens (and thus the same amount of training data) per training step. Thus, `XL:512` (segment length 512) runs at a batch size of 256 (8 per replica), while `Slide:12L` (segment length 4096) runs at a batch size of 32 (1 per replica) on PG19.

## 4.2   Benefit of Recurrence

We compare the 5 baselines to all six gate configurations for the Block-Recurrent Transformer. The recurrent model reliably outperforms all five baselines. The best overall configuration is `Rec:fixed:skip`, which outperforms the others in 3 out of 4 cases, and comes within the margin of error in the remaining case. This is especially notable because it is also the fastest configuration, having a slightly *lower* step time and fewer parameters than `Slide:13L`, because it does not have

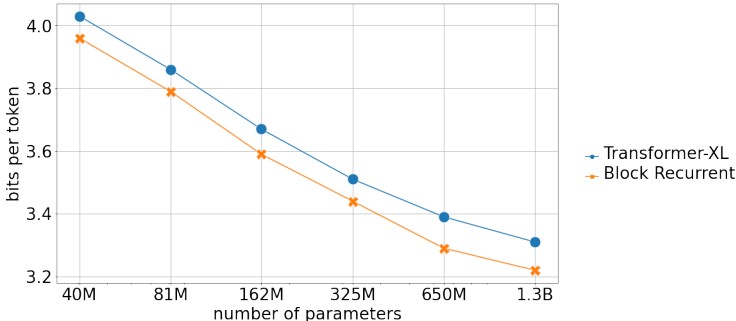

Figure 3: Scaling of the 12-layer Block-Recurrent Transformer vs 13-layer Transformer-XL on PG19. FLOPs are the same between the two models at a given parameter count. **At larger sizes, adding recurrence is equivalent to doubling the number of parameters.** Details in Appendix F.

the MLP. It is better than the 13-layer baseline by a wide margin, and it is even better than the Transformer-XL model with a window size of 2048, which runs over 2 times slower.

The other gate configurations also outperform the 13-layer baseline, but their relative ranking varies according to the dataset. Despite being theoretically more powerful, the `LSTM` gate tends to lag behind the `fixed` gate in all of our experiments.

**Scaling up.** Figure 3 shows the effect of adding recurrence as the transformer model is scaled up and down in size. We trained six different models on PG19, ranging in size from 40M parameters to 1.3B parameters. For the four smaller models, we compare a 12-layer Block-Recurrent Transformer against a 13-layer Transformer-XL baseline, while for the two larger models, we compare a 24-layer Block-Recurrent Transformer, with recurrence at layers 10 and 20, against a 26-layer Transformer-XL baseline. This experiment used a cosine-decay learning rate as described in [43], and a custom 32k SentencePiece vocabulary [44]. More details are in Appendix F.

Our experiments show that recurrence provides a consistent benefit across all scales. The relative improvement actually seems to increase with the number of parameters; at larger sizes **recurrence provides a benefit which is greater than doubling the number of parameters**.

### 4.3 Ablations

**Multiple recurrent layers.** Adding two recurrent layers right next to each other in the stack (layers 9 and 10) did not improve model perplexity. Adding two layers widely separated in the stack (layers 4 and 10) did provide an improvement, but the improvement was no better than simply adding another non-recurrent layer to the stack. Previous work on Memorizing Transformers [11] showed a similar effect. In our qualitative study, we saw that the model seems to use recurrence primary for long-range name lookups, much like memory. We conclude that one layer of recurrence is sufficient for the model to extract most of the benefits, although we did use two layers for our largest models.

**Number of recurrent state vectors.** We trained the model with differing numbers of state vectors, from 128 to 2048. Increasing the number of states makes a small but measurable improvement up to 1024, but the model does worse with 2048 (see Appendix D). We hypothesize that the model has trouble learning to use the recurrent state effectively if the state space grows too large.

**Reducing window size.** Reducing the size of the sliding window makes the perplexity significantly worse for Transformer-XL, because it reduces the amount of context that the transformer is able to attend to. Reducing the size of the window in a recurrent transformer has a smaller effect, because the model can use recurrence to compensate (see Appendix D).

### 4.4 Block feedback

Inspired by the feedback transformer [24], which allows all layers to attend to the topmost layer, we implemented a variation in which every layer of the transformer (not just the recurrent one) can cross-attend to the state vectors in the recurrent layer. This variation further improves perplexity, but

Table 2: Comparison with other published work on PG19. Fields marked - are unknown.

| Model | Layers | perplexity word-level | parameters | vocabulary size |
|---|---|---|---|---|
| Compressive Transformer [15] | 36 | 33.6 | - | 32k |
| Routing Transformer [7] | 22 | 33.2 | 490M[1] | 98k |
| Perceiver AR [45] | 60 | 28.9 | 974.6M[1] | 32k |
| Block-Recurrent Transformer | 24 | 28.46 | 650M | 32k |
| Block-Recurrent Transformer | 24 | **26.50** | 1.3B | 32k |

at a cost; step time increased by approximately 35-40%, and the additional queries also increase the number of parameters. Results are shown in Table 1, and further described in Appendix E.

## 4.5 Comparisons against prior published work

The PG19 test set contains 6,966,499 words [15], which are broken into 10,229,476 tokens using a SentencePiece vocabulary, trained on PG19. Our 24-layer 1.3B parameter model achieves 3.22 bits per token, and thus **achieves a new state of the art word-level perplexity of 26.50** (Table 2). However, we note that raw perplexity numbers are not necessarily a meaningful way to compare architectures, because they depend on numerous other factors, such as the number of parameters, vocabulary, learning rate schedule, batch size, etc.; a more detailed discussion is in Appendix C.3.

We were able to run a fair comparison (identical vocabulary, configuration, and hyperparameters) of the Block-Recurrent Transformer against the Memorizing Transformer [11], with a memory of size 64k (Table 1). The memorizing transformer is constructed similarly to our model; it has one layer which has been augmented with a mechanism that gives it the ability to attend over much longer distances. We find that Block-Recurrence does almost as well as the Memorizing Transformer on arXiv, and does just as well on PG19, but trains almost twice as fast. However, there are many ways of implementing approximate $k$-nearest-neighbor lookup, so relative speed will be highly implementation-dependent; our implementation runs on TPU, and does not use custom CUDA kernels.

## 4.6 Qualitative analysis

Prior work on long-context transformers [42, 11] has found that attention at long ranges is typically used to look up proper names, such as characters or places. We performed a qualitative analysis in an attempt to determine whether our model is using recurrence in the same way. We selected 5 books at random from the PG19 test set, ran both the Block-Recurrent Transformer and the 13-layer Transformer-XL on each book, and then compared the cross-entropy loss for all tokens. We sorted the results, and examined the top 4 tokens from each book with the greatest difference: the tokens for which the predictions of the recurrent model have the largest improvement over the baseline.

In 17/20 cases, the recurrent model predicted a proper name, usually with relatively high probability, that Transformer-XL was unable to predict. In 2 cases it predicted a chapter title (having previously seen the table of contents), and in the last case, it predicted a foreign-language word that was unique to that book. In 19/20 cases, the predicted word was nowhere within the attention window, so it must have been stored within the recurrent state (details in the appendix, Section G).

In a second study, we compared the recurrent model, running normally, against a variation in which the recurrent state is cleared at the end of each 4096-token segment, instead of being cached. Clearing the state degrades the model's ability to predict dependencies at a longer range than the segment length; typical mispredictions once again included proper names and chapter titles. Interestingly, this study also showed that the recurrent model is able to remember the title and author of a book (which is part of the Gutenberg boilerplate at the beginning and end of each book) **across the entire length of the book – more than 60,000 tokens.** See Appendix G.1.

A further quantitative comparison of the per-token cross-entropy between Transformer-XL and the Block-Recurrent Transformer is given in Appendix H.

---

[1]Personal communication.

# 5 Discussion

Our implementation of recurrence was inspired by the way that humans seem to process long sequences. When a human reads a novel, they do not attempt to remember every single word in the book. Instead, a human reader will construct a mental model, or knowledge graph, which summarizes the story thus far, i.e., the names of the main characters, the relationships between them, and any major plot points. When a human reads a paragraph of text, they will parse the information in the paragraph, process and interpret the information using background knowledge from their mental model, and finally update their mental model with new information. Our recurrent architecture loosely mimics this process. It takes a block of text, and parses it by running it through a conventional transformer stack. Tokens in the text attend to the recurrent states (i.e. the mental model), and the states, in turn, are updated by attending to the text.

Based on our qualitative analysis, it seems that the model is, in fact, using the recurrent state to summarize some of the information about frequently occurring characters and places. However, it does not seem to be doing much complex reasoning, as evidenced by the fact that our best performing model is the `fixed:skip` configuration. This configuration does not use a complex LSTM-style gate, which chooses to remember or forget based on its current state and inputs; instead, it simply computes an exponential moving average, not unlike some other forms of long-range approximate attention.

Moreover, the `skip` configuration cuts out the large MLP from the recurrent transformer layer. In a vanilla transformer, removing the MLP from all layers would severely degrade the model [46]; those large MLPs are computing something important. In a recurrent layer, removing the MLP makes little difference; it does not seem to be computing anything useful. We conclude that training the recurrent layer to make full use of its capabilities for knowledge extraction and summarization will require further advances.

## 5.1 Ethics

The potential negative social impacts from this work are similar to any other advance in language modelling. Large language models could potentially be used to create disinformation and fake news, power malicious chatbots, or generate spam. The Block-Recurrent Transformer can potentially create longer documents than was previously feasible, thus expanding the range of applications in which these negative impacts could occur. The best way to mitigate these risks is to train models that can reason about text, and flag misinformation or malicious content.

# 6 Conclusion

We have shown that when training language models on long documents, the Block-Recurrent Transformer provides a greater benefit at lower cost than scaling up the transformer model in other ways. Adding recurrence to a single layer has roughly the same cost as adding an additional non-recurrent layer, but results in a much larger improvement to perplexity. We have also shown that recurrence provides a larger benefit than simply increasing the window size of attention, or increasing the number of parameters. **Our medium-sized model has lower perplexity than a Transformer-XL model with 4 times the window size, but runs twice as fast, and our larger model outperforms a Transformer-XL model with twice the number of parameters**.

Furthermore, in contrast to some other recently proposed transformer variants, the Recurrent Transformer is very easy to implement, since it consists mostly of ordinary transformer components and RNN gates. No custom CUDA kernels are required. Our code has been released as open source [1].

Evaluating block-recurrent transformers on downstream tasks is an important direction for future work. We believe that the Block-Recurrent Transformer will be most useful in situations that require long-range context; examples of potential applications include writing book reports, summarizing long news articles, code completion, or question/answering over book-length works. There are are a number of new and emerging benchmarks that test long-range performance [47, 48, 4]. Previous studies have found a strong correlation between language modeling and diverse downstream tasks [49, 50].

Despite our initial successes, we also believe that the recurrent architecture that we present here has not yet achieved its full potential, and there are opportunities for future research and further improvements in this area.

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
