# OpenReview forum: "Block-Recurrent Transformers"
_NeurIPS.cc/2022/Conference — NeurIPS 2022 Accept_

### Official Review · Reviewer_o5rR · 2022-06-24

**Rating:** 6
**Confidence:** 5
**Soundness:** 3 good
**Presentation:** 2 fair
**Contribution:** 3 good

**Summary:**

The paper proposes a new model which adds block-wise outer recurrency to a Transformer.
The block-wise recurrency operates on a block of tokens, i.e. the hidden state is updated not after every single token but after a block of tokens.
The hidden state is not just a feature vector but a matrix, or a vector of features.
The Transformer accesses the hidden state by cross attention on the hidden state, i.e. those vectors of features.
The Transformer additionally uses sliding-window attention like the Longformer. At the beginning of the sequence, it has access to the previous sequence and also the previous hidden state via a cache, like the Transformer-XL and Longformer. The gradient flow obviously stops at the sequence boundary.
In training, one sequence of tokens consisting of multiple blocks is given to the model.
Going over the blocks and updating the recurrent state is orthogonal to going over the tokens. For any token, the hidden state it would access is the last hidden state it has access to.
The recurrency update is similar to the layerwise structure of a Transformer. Considering one Transformer layer including the cross attention to the hidden state, mostly the same structure is used for updating the hidden state. Specifically, it uses self-attention on the hidden state itself, and cross attention to the tokenwise features of the current block. However, instead of residual connections, they use a gating mechanism here. Thus, because there is the self-attention and cross-attention (already together) and then a following MLP, there are two gates per recurrent state update. They also have some variations on removing some of the Transformer components, like the MLP, such that the recurrent state update only has one gate. In their experiments, this simpler variant actually performs better.
This described actually only one single block-recurrent Transformer layer. Their overall model is actually mostly like Longformer, i.e. Transformer-XL with sliding-window attention, and they simply replace one single layer by such a block-recurrent Transformer layer.
They also test another variant (referred to as "Feedback" in the table) where they also modify the other standard Longformer layers by additional cross attention to the last recurrent state, where the recurrent state still only comes from a single layer.

The sliding-window attention is implemented also in a blockwise fashion for better efficiency in training. I think it actually uses the same blocks both for the recurrent state and the sliding-window attention, although this is a model aspect for the recurrency, and an implementation detail for the sliding-window attention.

They perform experiments on the PG19 dataset, arXiv dataset and GitHub dataset. They reach or surpass state-of-the-art performance in all.

**Questions:**

Memory Networks, Neural Turing Machines are related models considering the type of hidden state.

Perceiver also somewhat similar?


> Instead of processing the sequence one token at a time, our recurrent cell operates on blocks
> of tokens; see Figure 1. Within a block, all tokens are processed in parallel.

Processing refers to training or inference/generation? This should be made very clear. I assume this is about training because otherwise you cannot process tokens in parallel, right? At inference, even within a block, tokens are handled sequentially, I assume. This should be made clear. When it is not specifically said that it is about training, I assume inference and then this statement is wrong.

How is training implemented? A loop over the blocks?
How is inference implemented (or would be)?

I think, to make it really clear, the full definition of the model as it is used in inference (not training) should be given. Just a figure is not enough.
Recurrent cell is (mostly) clear, but the overall model is not clear. How is this recurrent cell integrated? How does the overall model look like?

> standard transformer layers ...
> We also normalize queries and keys

It would be good to clarify exactly how the overall model looks like. It seems like "standard Transformer layers" is not totally correct and there are differences, as mentioned later. Also, using sliding-window attention is also some big difference to the standard Transformer.

Is sliding-window attention and also the cache of the previous sequence also used for all the other Transformer layers?

It is not exactly clear: The self-attention in vertical direction, does it always get the last W tokens, i.e. potentially including the last block? Or is it always only within a block, so the first token would not get any context (except the recurrent state)? I assume always the last W tokens, but it's not exactly clear.

What is the effective context length in each case for each model? For the proposed recurrent case, it should be infinite, right? For the other variants, it is probably still quite high, sth like num layers * segment length but I'm not exactly sure. But this is an important aspect and should be discussed. For the Slide model variants, it should be less than the XL models, right?

> Appendix: Given a segment of N tokens (N = 4096 in our experiments), the sliding window applies a causal mask

How is this related to the blocks mentioned before?
Is this about self-attention? The self-attention is anyway only inside a block and not over the whole sequence?

In general (paper + appendix):

The terms "sequence" and "segment" also seem to be mixed up. This is confusing. The terminology should be consistent and it should be made clear what is meant.

Also the terms "window" and "block" seem to be mixed up. "blocks of length W", "window size W". I'm not exactly sure whether they refer to the same thing or not. Is there any reason it uses the same length variable W?

> Appendix: The sequence of N tokens is subdivided into blocks of size W

This is not the block concept from the block-recurrency, right? This is just an implementation detail here to make it more efficient, I assume. It's unfortunate that this causes some confusion here. This distinction should be made more clear. I think before implementation details for better efficiency are discussed, the model itself should be clearly defined.

> Appendix: The sliding window architecture has a context length of W for every token.

This statement is wrong when there are multiple layers. The context length is multiplied by the number of layers.Or if this is not the case, then something is very unclear here, and should be explained.

> current cell state at time t

What is time t? "Time" is highly confusing. This either means the state index (s in {1...S}) or the block index. I assume the block index but this is not clear here. So it means, the same gate is applied for the whole block? This should be clarified. And then also discussed. I'm not sure that this is the natural solution. Why not apply the gating per state index s?


Analysis:

Presentation in table 1 is a bit confusing. It's not explained well what the model notation means. From a first glance after not having carefully read the paper, it's not clear at all, which of those models actually correspond to the proposed block-recurrent transformer, etc. It becomes more clear after having carefully read everything, but still I think this is bad. Also, step time is unclear, is this for training or inference? PG19 perplexity on token-level is not so helpful - better would be on word-level, like it is also in table 2.

The Slide model (table 1), is this actually exactly a Longformer model? It would be better if this becomes clear.Or if the Slide model is not a Longformer, how exactly is it different? This is not clear then, and should be explained.


> Scaling up / table 2

It should be clarified how exactly the scaling is done. The dimension is increased?

Recurrent block size W, what is the effect? (Window size seems to be studied in the appendix, but not (recurrent) block size.)

I wonder about more extreme cases, like:

- S=1024, block size W=1, window size = 1, like NTM (Even though the trend in Appendix D might suggest this is bad, I think this is still interesting, whether it works at all, and how good this still is. Also, there might be a point where it improves again because it is forced to make more use of the recurrency also for short context, so it might learn that better.)
- S=1024, block size W=1, window size = 1024

(More recurrent layers? Discussed but not really systematically.)

Analysis on usefulness of recurrent state, depending on where you are in a block? Later on inside a block, the recurrent state might be less important?

Analysis on usefulness of self-attention in recurrent state update (horizontal direction). Is it actually used?

Analysis is quite short.

> We find that recurrence performs roughly on a par with the memorizing transformer, but trains considerably faster. We do not report the exact speedup ...

It would still be useful to get some impressions.

The Memorizing Transformer, I assume they have some own implementation because the numbers they report are not from the original paper. In the original paper, the best configuration used a segment/window length of 2048, and a memory size of 65k. Here they choose to use seg/win len 512 and mem size 32k. Why? This looks like they deliberately choose a suboptimal model such that their numbers look better in relation. Or if there is a reason, this should be explained, and then it would still be interesting to see the best Memorizing Transformer setting for sake of completeness.

Considering scaling the model up (Table 2), there the Memorizing Transformer would also be interesting in comparison.

After these results, I cannot tell whether the Memorizing Transformer is better or the Block-Recurrent Transformer.

> As with Transformer-XL, the keys and values from the last block are stored in a non-differentiable cache

This is the last block of the segment, which is for the next segment?
So this is actually different to Transformer-XL, where the whole segment is stored?
Or maybe I misunderstand sth here. This should be clarified.

> The baseline model is a 12-layer transformer

Where is this model in the table? Or is the baseline actually the XL model?

I wonder, why do the Slide models perform better than the XL models on PG19 but worse on Arxiv and Github? Because they have less context and context is more important on Arxiv or Github? This should be discussed.


**Limitations:**


It is mentioned that the full potential is likely not achieved yet and the current recurrent structure is maybe suboptimal.

Negative social impacts are also shortly but adequately addressed.


**Strengths And Weaknesses:**

Strengths:

The idea to combine recurrency with self-attention in some way is not novel and has often been tried before in various ways but often only with little success. From the results, it looks like the proposed model really improves considerably due to the recurrency. The results look good.

Overall, the authors do a good job in summarizing related work.

Plan to release the code.

Weaknesses:

In the next part under questions (suggestions), I will list more things individually. Here this is just a summary of the weaknesses.

Unfortunately the model definition is unclear in many parts. My summary is what I assumed after re-reading many parts again and again and inferring what would have made sense. But this is not good. The model definition should be completely unambiguous and very clear.

The analysis on aspects of the model is a bit short and leaves many open questions.

The experimental comparison to the Memorizing Transformer looks a bit unfair. It's not clear whether the Memorizing Transformer can yield better overall perplexity.

Some more standard language modeling benchmarks like enwik8 or Wiki-103 are not used. This would have been nice as there are more results from the literature to compare to.

Code not released yet. It's also not clear whether it really will be done. I have read statements like "we plan to" so often where it was never released in the end...
Also, in this case, as there were so many things unclear, the code could have helped a lot in clarifying everything exactly up to the latest detail.

---

> ### Author Response · Authors · 2022-08-02
> **Response to Reviewer o5rR:**
>
>
> Thank you for your valuable review!  We appreciate the time you spent giving us such detailed feedback.  We have updated the paper to clarify the points that you identified as confusing.  Most of our changes are to Section 3 (method).  With the increased page limit, we can now incorporate information that was previously in Appendix A into Section 3, and we have written a new version of Appendix A.  Changes to the paper are highlighted in blue.  Please take a look!
>
> Due to limits on response length, we cannot answer all of your questions here; we putting detailed answers to your questions in the supplementary material.  As requested, we have also released source code in the supplementary material.
>
> Since most of your feedback was on improving the presentation of our paper, we hope that you will consider increasing your review score in response to these changes.
>
> Response to weaknesses:
>
> > Code not released yet. It's also not clear whether it really will be done.
>
> The good news is that we released the code as open source on github more than a month ago!   We cannot share the link here without violating the double-blind review, but we will include a link in the camera-ready copy.  In the meantime, we are attaching a copy of the open source archive, which has been scrubbed of identifying information.  We totally agree that it would have been better to include the code with our original submission, since many of your questions might have been answered by looking at the code. [Done]
>
> > The experimental comparison to the Memorizing Transformer looks a bit unfair.
>
> The comparison uses a completely identical configuration between the Memorizing Transformer, and the Block-Recurrent transformer, so it is 100% fair.  The only difference is our use of recurrence in one layer, vs their use of Top-k memory in one layer.  The purpose of the comparison is to see whether recurrence provides performance comparable to Top-k memory.  Both the block-recurrent transformer and the memory transformer can be scaled up, both in terms of the number of parameters and the window size, but that's not the point.
>
> We did use a somewhat smaller memory size than the original paper, but we're happy to re-run that comparison with a larger memory if you like.  It won't make much difference.  Even with the smaller size, our numbers closely match the numbers reported in the Memorizing Transformer paper.  We report numbers in bits-per-token, so you have to convert; they get 3.54/1.21 bpt for pg19/arxiv, while we report 3.50/1.24.
>
> > Some more standard language modeling benchmarks like enwik8 or Wiki-103 are not used.
> > This would have been nice as there are more results from the literature to compare to.
>
> We agree that having standard benchmarks is useful for comparing architectures, and enwik8 has been historically used in many papers for long-range modeling. However, in our opinion, it's not a particularly good benchmark, at least for our purposes.
>
> The purpose of our experiments is to see whether block recurrence can transmit information over very long lengths: we show retrieval over 60k+ tokens.  We chose PG19 specifically because we believe it to be a good dataset for these sorts of experiments.  It consists only of long, book-length works, it is much larger than enwik8, it is publicly available, and has been cited in other published work.  Arxiv and github are (sadly) not public, but they similarly have long documents in the 50k+ token range.
>
> Enwik8 is not a corpus of long articles.  In fact, it doesn't even split the text into separate articles at all;  it's just a single text dump that concatenates a bunch of short unrelated articles together.  If you do attempt to split it, you will discover that the majority of "articles" are merely stubs, with HTML boilerplate and no actual text.  Enwik8 is a fine benchmark for data compression, which was the purpose for which it was originally intended, but it is less than ideal for long-range language modeling.
>
> Wiki-103 is better because it does break the text into articles, and it eliminates the boilerplate, but the average length is still only 3.6k tokens per article, which is less than the segment length used in our experiments, and a far cry from the 50k+ tokens per book of PG19.
>
> Note that unlike many other papers, our architecture does not incorporate any "special tricks" to improve short-range language modeling, so we would expect block-recurrence to provide at best a modest improvement over the Transformer-XL baseline on these data sets.
>
> *** Answers to your other questions are in the supplementary material. ***

---

> > ### Author Response · Authors · 2022-08-03
> > **Response to Reviewer o5rR, Part II**
> >
> >
> > In the interest of making this discussion more visible, we decided to post answers here, using multiple responses, rather than putting them in the supplementary material.
> >
> > > The Slide model (table 1), is this actually exactly a Longformer model? It would be better if this becomes clear. Or if the Slide model is not a Longformer, how exactly is it different?
> >
> > The Slide model is not a Longformer; it only implements the sliding-window attention pattern.  Longformer is a much more complicated model, and sliding-window attention is merely one of several different attention patterns that it uses.  In addition to the sliding window, Longformer also uses dilated attention and sparse global attention, both of which are implemented with custom CUDA kernels.  Moreover, LongFormer uses different window sizes in each layer, and it uses a multi-phase training regimen of pre-training and fine-tuning, following a curriculum that gradually increases window size and sequence length in order to achieve their final results.  We do none of these things.  (As a side note, Longformer is an excellent example of what we mean by "special tricks".)
> >
> > >> "Instead of processing the sequence one token at a time, our recurrent cell operates on
> > >> blocks of tokens; see Figure 1. Within a block, all tokens are processed in parallel."
> > > Processing refers to training or inference/generation? This should be made very clear. I
> > > assume this is about training because otherwise you cannot process tokens in parallel, right?
> >
> > You are absolutely correct, and that is an excellent point.  Since this is an autoregressive model, inference must still be done sequentially, token by token.  The benefit of handling blocks of tokens in parallel only happens during training.  We have clarified this point in the paper.  [Done]
> >
> > > How is training implemented? A loop over the blocks? How is inference implemented (or would be)?
> >
> > Training is a loop over the blocks; all tokens within a block are processed in parallel.  Inference is a nested loop: an outer loop over the blocks (just like training) and an inner loop over the tokens within each block (for autoregressive decoding).  The inner loop is the same as a vanilla transformer.  We have updated the paper.  [Done]
> >
> > > The terms "sequence" and "segment" seem to be mixed up. This is confusing. The
> > > terminology should be consistent and it should be made clear what is meant.
> >
> > We have tried very hard not to mix up these terms.  A "sequence" refers to all of the tokens within a document; in the case of PG19, it refers to all of the tokens within a book.  You are right that this term was not properly defined, and we have updated the paper to clarify this point. [Done]
> >
> > The word "segment" is defined in Section 3 (Method):  "we divide the document [i.e. the full book-length sequence] up into segments of length N, which are processed sequentially over a number of training steps. Each training step processes one segment. Each segment is further divided into blocks of length W."
> >
> > The confusion between sequence/segment may be caused by the fact that most research on transformers does not distinguish between the two.  The typical way to train a transformer is to split a document up into segments, and then shuffle them, so the model never sees the original full-length document.  In that case, "sequence" and "segment" are the same thing.  Thus, in section 2 (related work), we simply use the term "sequence".
> >
> > > Also the terms "window" and "block" seem to be mixed up. "blocks of length W", "window size
> > > W". I'm not exactly sure whether they refer to the same thing or not. Is there any reason it
> > > uses the same length variable W?
> >
> > Unlike sequence/segment, window/block mean very similar things, and are used somewhat interchangeably.  We agree that this is confusing, and have clarified this in the paper; thanks for pointing that out.  Part of the problem here is that due to the 9-page limit, we moved the section describing sliding window attention into Appendix A, where readers do not see it.  We have used the increase in page limit to move this information back to its original position, in Section 3 (Method), which will hopefully be clearer.  [Done]
> >
> > The term "window" comes from "sliding window attention", which is a term of art that was introduced previously in the literature, and refers to the local distance over which each token can attend within a single layer.  We divide each segment into blocks, and both the sliding window and recurrence are implemented over blocks, so the block size and the window size are the same.
> >
> > From the (original) Appendix A:  "The sliding window applies a causal mask in which each token can only attend to the W previous tokens, where W is the window size... [sliding window attention is implemented as follows] ... The segment of N tokens is subdivided into blocks of size W, and each block attends locally to itself and to the previous block, so the size of each local attention matrix is W × 2W".

---

> > > ### Author Response · Authors · 2022-08-03
> > > **Response to Reviewer o5rR, Part III**
> > >
> > >
> > > >> Appendix: "Given a segment of N tokens (N = 4096 in our experiments), the sliding window
> > > >> applies a causal mask"
> > > > How is this related to the blocks mentioned before? Is this about self-attention? The
> > > > self-attention is anyway only inside a block and not over the whole sequence?
> > >
> > > Yes, this is about self-attention.  Mathematically, self-attention is defined over the entire sequence (not just the segment), no matter how long.  However, if there is a causal mask such that each token can only refer to the W previous tokens (e.g. a sliding-window mask), then most of that massive attention matrix will be filled with zeros, as illustrated in Figure 2.  (Previously Figure 3.)  Thus, as an optimization, the sequence can be divided into segments, the segments are divided into blocks (of size W), and attention is computed locally within pairs of neighboring blocks.
> > >
> > > >> Appendix: "The sequence of N tokens is subdivided into blocks of size W"
> > > > This is not the block concept from the block-recurrency, right?  This is just an implementation
> > > > detail here to make it more efficient...
> > >
> > > It is the same block concept.  The blocks are an implementation detail in non-recurrent layers, but they are more than just a detail in the recurrent layer.  We use the same blocks to implement both sliding-window attention and recurrence.  The block-recurrent cell of Figure 1 handles both self-attention and recurrence.  Please take a look at the newly updated Section 3, which should hopefully make this clearer.
> > >
> > > > It is not exactly clear: The self-attention in vertical direction, does it always get the last W
> > > > tokens, i.e. potentially including the last block?
> > >
> > > Yes.  As described in the (original) Appendix A, every token can attend to W previous tokens, which means that it can attend to tokens within the current block, and within the previous block.  For the first block in a segment, the "previous block" is the (cached) last block from the previous segment.  We've clarified this point.  [Done]
> > >
> > > >> "As with Transformer-XL, the keys and values from the last block are stored in a non-differentiable cache."
> > > > This is the last block of the segment, which is for the next segment? So this is actually
> > > > different to Transformer-XL, where the whole segment is stored?
> > >
> > > You are correct.  Transformer-XL does not use a sliding window, so the segment length and the window/block length are the same (N=W).  In TXL, caching the last "block" thus caches the entire segment.  The sliding window architecture allows the segment length to be longer than the block length (N >> W), but attention still can't look past a single block, so only the last block needs to be cached.  We've clarified the paper.  [Done]
> > >
> > > > Is sliding-window attention and also the cache of the previous sequence also used for all the
> > > > other Transformer layers?
> > >
> > > Good question!  Yes.  Sliding-window attention is used in all layers.  We have clarified that point. [Done]
> > >
> > > >> "Appendix: The sliding window architecture has a context length of W for every token."
> > > > This statement is wrong when there are multiple layers. The context length is multiplied by the
> > > > number of layers.Or if this is not the case, then something is very unclear here, and should be
> > > > explained.
> > >
> > > You're right, but we mentioned the number of layers in the very next paragraph!  From the (original) Appendix A:  "Thus, the theoretical receptive field (the maximum distance that information can propagate through the model) [of sliding attention] is W * L, where L is the number of layers."
> > >
> > > The purpose of that particular section is to distinguish between the "context length", which we defined as the distance over which tokens can attend within a single layer, and the "theoretical receptive field," which applies to the model as a whole.   (Note that the term "receptive field" is also used in the longformer paper; we did not invent this terminology.)

---

> > > > ### Author Response · Authors · 2022-08-03
> > > > **Response to Reviewer o5rR, Part IV**
> > > >
> > > >
> > > > > What is the effective context length in each case for each model? For the proposed recurrent
> > > > > case, it should be infinite, right? For the other variants, it is probably still quite high, sth like
> > > > > num layers * segment length but I'm not exactly sure. But this is an important aspect and
> > > > > should be discussed. For the Slide model variants, it should be less than the XL models,
> > > > > right?
> > > >
> > > > We use the term "theoretical receptive field" (TRF) rather than "effective context length", because there is no guarantee that the model will actually be able to use the whole TRF in practice.  For example, the TRF for an LSTM is theoretically infinite, but in practice LSTMs tend to forget after a few hundred tokens.  Please take a look at the newly written Appendix A, which now covers this subject in more detail.
> > > >
> > > > The Slide model and Transformer-XL have the same TRF, but Slide makes better use of it.
> > > > Section 4.1: "XL:512 ... uses a transformer-XL style cache, but no sliding window, so the segment length is the same as the window size, i.e., it is trained on segments of 512 tokens."
> > > > Section 4.1: "Slide:12L is a 12-layer transformer with a window size of 512, but uses a sliding window over a segment of 4096 tokens. This model is almost identical to XL:512; the only difference is that the sliding window is differentiable over multiple blocks, while the Transformer-XL cache is not."
> > > >
> > > > In other words, both XL:512 and Slide:12L have a window/block size of 512, and a cache of size 512.  The XL model has a segment length of 512, but the Slide model has a segment length of 4096, divided into blocks of size 512.
> > > > The TRF for both of these models is the same: W * L, but the Slide model can propagate gradients along the entire segment length, while the XL:512 model cannot.  The XL:512 model can attend into the cache, but cannot propagate gradients through it, because the cache is not differentiable.
> > > >
> > > > In both cases, making use of the full TRF of W * L would require L separate "hops" of attention, each of which would have to attend the maximum distance, which is unlikely in practice.  If one defines "effective context length" as "the amount of context that the model actually learns to use", then we would expect the "effective context length" to be slightly higher in the Slide model, because it is more differentiable.  Our experiments back this up: Slide:12L does have slightly better perplexity than XL:512.
> > > >
> > > > The TRF for the recurrent model is infinite, and in our qualitative studies, we show that the "effective context length" seems to be very long in practice; we show that it can look up names over distances of more than 60k tokens.  This is a huge increase; the Slide model has a TRF of only 6k tokens, and an effective context length shorter than that.
> > > >
> > > > >> "current cell state at time t"
> > > > > What is time t? "Time" is highly confusing. This either means the state index (s in {1...S}) or
> > > > > the block index. I assume the block index but this is not clear here. So it means, the same
> > > > > gate is applied for the whole block? This should be clarified. And then also discussed. I'm not
> > > > > sure that this is the natural solution.
> > > >
> > > > We agree that "time" is not a great word, because there is no physical concept of time.  However, "time" has traditionally been used to describe successive iterations of a recurrent architecture, as in "backpropagation through time."  In this case it means block index, because recurrence is at the block level.   We have updated the paper to clarify this point. [Done]
> > > >
> > > > > Why not apply the gating per state index s?
> > > >
> > > > As mentioned in the introduction, all state vectors are processed in parallel.  That's more or less the whole point of the architecture!  Using the state index for gating would make no sense; if we were to process states sequentially, then "block-recurrence" would be a weird operation with no advantages over traditional per-token recurrence.
> > > >
> > > > As mentioned in Section 3 (Method), gates replace the residual connections in a standard transformer layer.  Thus, there is a separate gate for each state vector, but all state vectors are updated in parallel.  We have added a sentence to clarify this point.  [Done]
> > > >
> > > > > Presentation in table 1 is a bit confusing. It's not explained well what the model notation means.
> > > >
> > > > The 5 baselines are described by name in section 4.1, immediately below the table.  The gate type and configuration are described in previous sections.  We have added a sentence to the caption which explains that "rec:gate:config" is the recurrent architecture.  Does that help?  [Done]
> > > >
> > > > >> "The baseline model is a 12-layer transformer"
> > > > > Where is this model in the table? Or is the baseline actually the XL model?
> > > >
> > > > That's the first baseline: XL:512.  All of the other baselines and recurrent models are simply variations on that architecture.  We have clarified the paper.  [Done]

---

> > > > > ### Author Response · Authors · 2022-08-03
> > > > > **Response to Reviewer o5rR, Part V**
> > > > >
> > > > > > [Presentation in table 1] Also, step time is unclear, is this for training or inference?
> > > > >
> > > > > As mentioned in section 4.1 (relative cost), it is for training.  We've updated the caption to clarify this point.  [Done]
> > > > >
> > > > > Inference time is also virtually identical between the block-recurrent architecture and transformer XL.  Recurrence introduces an extra attention operation over states, which must be done for each token, but only in one layer, and the 13-layer Slide baseline adds an extra layer to compensate.   We've updated the paper.  [Done]
> > > > >
> > > > > > The Memorizing Transformer, I assume they have some own implementation because the
> > > > > > numbers they report are not from the original paper. In the original paper, the best
> > > > > > configuration used a segment/window length of 2048, and a memory size of 65k.
> > > > > > Here they choose to use seg/win len 512 and mem size 32k.
> > > > >
> > > > > We are using the Memorizing Transformer code from the original paper, but we report numbers in bits per token.  If you convert, the original paper reports 3.54/1.21 bpt for pg19/arxiv, while we report 3.50/1.24, so our numbers are quite close to theirs.  The difference on arxiv is most likely due to the fact that we use a lower learning rate (Appendix C, training details).  The Memorizing Transformer does not use sliding window attention, so the segment and window length are the same (just like Transformer-XL).  We set the window length to 512 in both models in order to make a completely apples-to-apples comparison between memory and recurrence.
> > > > >
> > > > > The choice of 32k memory size is admittedly a bit arbitrary; we can easily rerun those numbers with a 64k memory instead if you would prefer.  There's not much difference between 32k/64k (notice that we actually got a better result on PG19 with 32k than they did with 64k).
> > > > >
> > > > > > After these results, I cannot tell whether the Memorizing Transformer is better or the
> > > > > > Block-Recurrent Transformer.
> > > > >
> > > > > The memorizing transformer tends to be a bit better in terms of perplexity.  Based on qualitative studies, both memory and recurrence seem to be used mainly for long-range name lookups, and it's probably easier for the model to do name lookups using top-k memory than to store names in recurrent states.  However, the two are surprisingly close, and recurrence is both faster to train, and has faster inference.
> > > > >
> > > > > There's also a qualitative difference between the two.  We hypothesize that memory is better for retrieving fine-grained details, while the recurrent states compress and summarize the history.  Recurrence may be better at capturing high-level summaries, or more nebulous characteristics like writing style, while memory is better at looking up facts.  More research is needed to characterize which one is "better" on downstream tasks.
> > > > >
> > > > > >> "We find that recurrence performs roughly on a par with the memorizing transformer, but
> > > > > >> trains considerably faster. We do not report the exact speedup ..."
> > > > > > It would still be useful to get some impressions.
> > > > >
> > > > > The block-recurrent transformer is almost twice as fast to train as the Memorizing Transformer.  We've update the paper.  [Done]
> > > > >
> > > > > > It should be clarified how exactly the scaling is done. The dimension is increased?
> > > > >
> > > > > Yes, the various dimensions are increased in factors of 2, including the number of layers for the largest models.  Configurations for various sizes are in the open source release.   We've added more details in Appendix F.  [Done]
> > > > >
> > > > > > I wonder about more extreme cases, like: S=1024, block size W=1, window size = 1, like NTM
> > > > >
> > > > > In our current code, this would also set the length of the attention window to 1, so it would no longer be a transformer.  :-)  It would also be very expensive to run.
> > > > >
> > > > > > Analysis on usefulness of self-attention in recurrent state update (horizontal direction). Is it
> > > > > > actually used?
> > > > >
> > > > > That's a really good question!  I'm not sure we have time to do that analysis in the upcoming week, but it would be a good thing to discuss for the camera-ready copy.
> > > > >
> > > > > > I wonder, why do the Slide models perform better than the XL models on PG19 but worse on
> > > > > > Arxiv and Github? Because they have less context and context is more important on Arxiv or
> > > > > > Github? This should be discussed.
> > > > >
> > > > > That's another really good question.  Slide:12L always outperforms XL:512, which has the same window length, because it's more differentiable.  The surprising thing is that on PG19,  Slide:12L (W=512) actually matches the XL:2048 model, which has a much longer window length, but it doesn't on arxiv or github.  Our guess is that arxiv and github have a lot of complex syntax and name lookups, and thus get a lot of benefit from being able to do direct attention (i.e. single-hop lookups) to matching tokens.  PG19 is natural language, and thus has fewer direct dependencies in general, but may benefit from more subtle dependencies that use multi-hop attention lookups.  Multi-hop lookups are easier to train in the slide model, because it is fully differentiable over the 4k sequence length.

---

> > > > > > ### Comment · Reviewer_o5rR · 2022-08-03
> > > > > > **Response**
> > > > > >
> > > > > > Thank you for the very detailed response, and for the updates! Also thanks for releasing the code!
> > > > > >
> > > > > > Some of your answers did not result in changes in the paper, while I think it would be helpful for other readers. E.g.:
> > > > > >
> > > > > > - Memorizing Transformer comparison.
> > > > > >
> > > > > > - Discussion on other datasets (enwik8 etc). Note that I still think that Wiki-103 would have been good for comparison.
> > > > > >
> > > > > > - Difference of Slide model vs Longformer (very briefly).
> > > > > >
> > > > > > ---
> > > > > >
> > > > > > > Appendix: "The sequence of N tokens is subdivided into blocks of size W"
> > > > > >
> > > > > > "Sequence" should be "segment", right?
> > > > > >
> > > > > > ---
> > > > > >
> > > > > > >> Also the terms "window" and "block" seem to be mixed up. "blocks of length W", "window size W".
> > > > > > >
> > > > > > > ... window/block mean very similar things, and are used somewhat interchangeably.
> > > > > >
> > > > > > But they don't need to be the same length, right? Except that this probably simplifies the implementation.
> > > > > >
> > > > > > Given that they don't need to be the same, I wonder if they really should be the same, or how it would perform if the sliding window is smaller or larger than the block size.
> > > > > >
> > > > > > ---
> > > > > >
> > > > > > I think a small sentence saying that the theoretical receptive field (TRF) is infinite for the Block-recurrent Transformer would help.
> > > > > >
> > > > > > ---
> > > > > >
> > > > > > > Thus, there is a separate gate for each state vector, but all state vectors are updated in parallel.
> > > > > >
> > > > > > This is still not perfectly clear. It would probably help to specify the dimensions e.g. of the h tensor. h_t is of shape (batch-dim excluded) [S,D], and thus gate i_t and f_t are of the same shape [S,D]? But for the fixed gate case, g just has shape [D]?
> > > > > >
> > > > > > ---
> > > > > >
> > > > > > >> I wonder about more extreme cases, like: S=1024, block size W=1, window size = 1, like NTM
> > > > > > >
> > > > > > > In our current code, this would also set the length of the attention window to 1, so it would no longer be a transformer.  :-)  It would also be very expensive to run.
> > > > > >
> > > > > > Yes, that's the point. It would be a transition between a Transformer and a more traditional LSTM, however, due to having a larger S>1, it would have a memory component similar to a NTM.
> > > > > >
> > > > > > The model formulation and these hyper parameter allow for such transition, so it would be interesting to directly compare it.
> > > > > >
> > > > > > Having S=1 and W=1 is maybe also interesting. This should be just as fast as a normal recurrent net.
> > > > > >
> > > > > > ---
> > > > > >
> > > > > > I think it's a bit unfortunate that many things are left to the appendix.

---

> > > > > > > ### Author Response · Authors · 2022-08-04
> > > > > > > **Re: Reponse (followup to Reviewer o5rR)**
> > > > > > >
> > > > > > >
> > > > > > > Thanks for taking a second look!  We're glad our clarifications helped.
> > > > > > >
> > > > > > > > Some of your answers did not result in changes in the paper, while I think it would be helpful for other
> > > > > > > > readers. E.g.:
> > > > > > > > * Memorizing Transformer comparison.
> > > > > > > > * Discussion on other datasets (enwik8 etc). Note that I still think that Wiki-103 would have been good for comparison.
> > > > > > > > * Difference of Slide model vs Longformer (very briefly).
> > > > > > >
> > > > > > > We're happy to add the discussion of datasets, and the longformer comparison to the appendix; that's not a problem.
> > > > > > >
> > > > > > > However, we're not sure that there's much more to say about the Memorizing Transformer at this point that isn't speculation; a scientific comparison will require follow-up experiments on downstream tasks.  We can put the speculation in the appendix if you want.  :-)
> > > > > > >
> > > > > > > BTW, wrt. to the Memorizing Transformer, we finished re-running the experiments with 64k memory, as you requested.  The numbers are very close.  With the additional memory, the Memorizing Transformer does slightly better than Block-Recurrence on arXiv, but two are exactly the same on PG19.   We have updated the paper accordingly.
> > > > > > >
> > > > > > > Unfortunately, there's not enough time left during the rebuttal period to try to run experiments with Wiki-103, since we don't have a data pipeline for Wiki-103 set up.
> > > > > > >
> > > > > > > > Appendix: "The sequence of N tokens is subdivided into blocks of size W"
> > > > > > > > "Sequence" should be "segment", right?
> > > > > > >
> > > > > > > You're right.  Fixed.  [Done]
> > > > > > >
> > > > > > > > I think a small sentence saying that the theoretical receptive field (TRF) is infinite for the
> > > > > > > > Block-recurrent Transformer would help.
> > > > > > >
> > > > > > > Absolutely.  We already added that to the new Appendix A, as per your earlier suggestion.  [Done]
> > > > > > >
> > > > > > > >> Also the terms "window" and "block" seem to be mixed up. "blocks of length W", "window size W"
> > > > > > > >>... window/block mean very similar things, and are used somewhat interchangeably.
> > > > > > >
> > > > > > > > But they don't need to be the same length, right? Except that this probably simplifies the implementation.
> > > > > > > > Given that they don't need to be the same, I wonder if they really should be the same, or how it would perform if the
> > > > > > > > sliding window is smaller or larger than the block size.
> > > > > > >
> > > > > > > Mathematically, the window size and the block length can be defined separately, but in terms of implementation, it is very convenient to keep them the same.  The window size can easily be made smaller than the recurrent block length by changing the causal mask, but there's no benefit to doing so.  Making the window size larger would significantly complicate matters, because then the window-blocks would have to be further subdivided into recurrent-blocks, and self-attention would have to be done outside of the recurrent cell.  Our current code doesn't support that option.
> > > > > > >
> > > > > > > > > Thus, there is a separate gate for each state vector, but all state vectors are updated in parallel.
> > > > > > >
> > > > > > > > This is still not perfectly clear. It would probably help to specify the dimensions e.g. of the h tensor. h_t is of shape
> > > > > > > > (batch-dim excluded) [S,D], and thus gate i_t and f_t are of the same shape [S,D]? But for the fixed gate case, g
> > > > > > > > just has shape [D]?
> > > > > > >
> > > > > > > Yes, g has shape [D], broadcast over [S, D]; it acts exactly like the bias terms in the gate equations, which also have dimension [D], broadcast over [S,D].   We will update the paper to clarify that.  (Come to think of it, using separate per-state values for g might be an interesting ablation.)
> > > > > > >
> > > > > > > >>> I wonder about more extreme cases, like: S=1024, block size W=1, window size = 1, like NTM
> > > > > > >
> > > > > > > >> In our current code, this would also set the length of the attention window to 1, so it would no longer be a
> > > > > > > >> transformer. :-) It would also be very expensive to run.
> > > > > > >
> > > > > > > > Yes, that's the point. It would be a transition between a Transformer and a more traditional
> > > > > > > > LSTM, however, due to having a larger S>1, it would have a memory component similar to a
> > > > > > > > NTM.  The model formulation and these hyper parameter allow for such transition, so it
> > > > > > > > would be interesting to directly compare it.
> > > > > > >
> > > > > > > That's an interesting idea, but at W=1, the block-recurrent model would run a whole transformer layer for each token, which would be very expensive; it would essentially be a 512-layer transformer (or worse, a 4096-layer transformer!).  I think our current implementation would very quickly run out of device memory without some clever gradient checkpointing hacks.
> > > > > > >
> > > > > > > > Having S=1 and W=1 is maybe also interesting. This should be just as fast as a normal recurrent net.
> > > > > > >
> > > > > > > You would still be computing the (now useless) key,value,queries, plus a projection, plus a large MLP, so it would be more expensive than an LSTM, although the 2-layer MLP might provide some benefit.
> > > > > > >
> > > > > > > > I think it's a bit unfortunate that many things are left to the appendix.
> > > > > > >
> > > > > > > Indeed!  However, due to page limits, we can't fit everything in the main text.  It's always difficult to figure out what to cut...

---

> > > > > > ### Author Response · Authors · 2022-08-05
> > > > > > **Update to paper & appendix.**
> > > > > >
> > > > > >
> > > > > > Minor update to paper (64k Memorizing Transformer numbers, wording changes).
> > > > > > New sub-sections added to supplementary material, Appendix A, to address reviewer concerns.

---

### Official Review · Reviewer_iE1S · 2022-07-11

**Rating:** 5
**Confidence:** 3
**Soundness:** 3 good
**Presentation:** 3 good
**Contribution:** 3 good

**Summary:**

This paper is interesting. The authors propose a new RNN structure for transformer by introducing new transition states between transfomers and acheived new SOTAs.

**Questions:**

1. Can this network be applied to other NLP tasks ?

**Limitations:**

1. Some failure cases can be given in the paper.

**Strengths And Weaknesses:**

Strengths
1. New SOTA results by cascading many transfomers with trival modifications.
2. Tackled the bottleneck the transformer by using RNN structure for language modelling.
Weaknesses
1. This network may be more effective for very long sequences.

---

> ### Author Response · Authors · 2022-08-02
> **Response to reviewer iE1S:**
>
>
> Thank you for your feedback!  We have responded to your comments below, and also updated the paper. Please let us know if your concerns are sufficiently addressed, or if you have any other questions.
>
> > Can this network be applied to other NLP tasks ?
>
> Yes, but the block-recurrent transformer will be primarily useful in situations that require long-range context, such as writing book reports, summarizing long news articles, code completion, question/answering over book-length works, or chat bots that require a long chat context.  These applications remain as future work.  We have updated the paper, and added this information to the conclusion.  [Done]
>
> > Some failure cases can be given in the paper.
>
> There are two failure cases that we discuss in the paper.  First, there is a failure mode where the transformer learns to ignore the recurrent state; we provide details on gate initialization to avoid this failure mode.  Second, as we discuss in Section 5 (Discussion), the block-recurrent transformer does not seem to be making full use of the capabilities of the LSTM gate.  Thus, we believe that the recurrent architecture that we present here has not yet achieved its full potential, and there are opportunities for future research and further improvements in this area.

---

### Official Review · Reviewer_aK9j · 2022-07-11

**Rating:** 7
**Confidence:** 4
**Soundness:** 3 good
**Presentation:** 3 good
**Contribution:** 4 excellent

**Summary:**

This paper presents a new architecture called the Block-Recurrent Transformer. Input sequences are chunked as a block, and each block is operated by a transformer layer. Each block is connected with a recurrent layer. Block id encoding similar to position encoding is introduced. The authors include ablation studies on different variants of the gate mechanism in the recurrent layer.

**Questions:**

Can we extend the method to a bidirectional recurrent layer?

Why is adding only one recurrent layer enough?

Can you formalize the complexity by a function of N, W, and S?


**Limitations:**

The paper does not explain how we can use the Block-Recurrent Transformer for a wide range of applications.

**Strengths And Weaknesses:**

The paper is well-written, and the motivation is clear. The Block-Recurrent Transformer improves the efficiency-accuracy trade-off compared to the Transformer-XL, which is a strong language model baseline. Here, the efficiency is the number of parameters and runtime, and the accuracy is language modeling perplexity. It would be much better to show that the Block-Recurrent Transformer could also be effective in many applications where Transformers are successful. However, the new architecture is only tested on language modeling.

---

> ### Author Response · Authors · 2022-08-02
> **Response to Reviewer aK9j:**
>
>
> Thank you for your encouraging review!
>
> > Can we extend the method to a bidirectional recurrent layer?
>
> Yes, but there are caveats.  A bidirectional layer obviously can't be used for autoregressive language modeling, because it violates causality, so you would have to use some other training objective, like masked language modeling.  Our research has also been on long, book-length works, and each book is broken into multiple segments during training.  Only one segment will fit into device memory at a time.  Running a bidirectional layer over an entire book would thus require two passes, one for each direction, and would be quite expensive.
> A bidirectional layer where the reverse direction only operates within a single segment, (thus providing look-ahead only within that segment), would be easy to implement.
>
> > Why is adding only one recurrent layer enough?
>
> Good question!  Based on our qualitative experiments, the recurrent layer seems to be most useful for long-range name lookups, like proper names and places in a novel.  A single layer seems to provide sufficient long-range memory to keep track of the proper names within a book, so adding more recurrent layers doesn't necessarily add more capability.  Related work on Memory Transformers have found a similar effect.
> In contrast, the parameters in the non-recurrent transformer layer are used to hold common-sense information about word meanings and general world knowledge, so more layers (more parameters) are always better.
>
> We've updated the paper to clarify this point. [Done]
>
> > Can you formalize the complexity by a function of N, W, and S?
>
> For the recurrent layer:
> (N/W) * (W^2 + S^2 + 2SW), where N is the segment length, W is the block/window size, and S is the number of states.  N/W is the number of blocks; and each block does self attention (W^2), state self-attention (S^2), and attention between tokens/states and states/tokens (2SW).  Note that in our experiments, we set S=W for simplicity, so this reduces to N*W.
> For the non-recurrent layers:
> (N/W) * W^2 = N*W
>
> We have added this equation to the paper.  [Done -- Appendix A.]
>
> > (Limitations): The paper does not explain how we can use the Block-Recurrent Transformer for a wide range of applications.
>
> The block-recurrent transformer will be primarily useful in situations that require long-range context, such as writing book reports, summarizing long news articles, code completion, question/answering over book-length works, or chat bots that require a long chat context.  These applications remain as future work.  We have updated the paper, and added this information to the conclusion.  [Done]

---

### Meta-Review · Area_Chair_CM2y · 2022-08-23

**Recommendation:** Accept
**Confidence:** Certain

**Metareview:**

This paper describes a modification to the transformer architecture to use block-recurrence to more accurately model very long sequences, borrowing some ideas from the LSTM. The idea is fairly simple to implement, as it doesn't require much code over a traditional transformer, and results seem good, if not completely overwhelmingly so.

All reviewers voted to accept this paper and I agree. It's a fairly simple idea with fairly good results and adds to the body of knowledge regarding how to model very long sequences.

**Award:**

No

---

### Decision · Program_Chairs · 2022-09-14

Accept